# RETINA: Reconstruction-based pre-trained enhanced TransUNet for electron microscopy segmentation on the CEM500K dataset

**Cheng Xing** [ID][1,2], **Ronald Xie**[1,2], **Gary D. Bader** [ID][1,2,3,4,5,6]*

**1** Department of Molecular Genetics, University of Toronto, Toronto, Ontario, Canada, **2** The Donnelly Centre, University of Toronto, Toronto, Ontario, Canada, **3** Department of Computer Science, University of Toronto, Toronto, Ontario, Canada, **4** The Lunenfeld-Tanenbaum Research Institute, Sinai Health System, Toronto, Ontario, Canada, **5** Princess Margaret Cancer Centre, University Health Network, Toronto, Ontario, Canada, **6** CIFAR Multiscale Human Program, CIFAR, Toronto, Ontario, Canada

\* gary.bader@utoronto.ca

**Data availability statement:** The benchmark datasets, the pre-trained model weights, and the full code of RETINA implementation are available on GitHub at the following link: https://github.com/BaderLab/RETINA.

## Abstract

Electron microscopy (EM) has revolutionized our understanding of cellular structures at the nanoscale. Accurate image segmentation is required for analyzing EM images. While manual segmentation is reliable, it is labor-intensive, incentivizing the development of automated segmentation methods. Although deep learning-based segmentation has demonstrated expert-level performance, it lacks generalizable performance across diverse EM datasets. Current approaches usually use either convolutional or transformer-based neural networks for image feature extraction. We developed the RETINA method, which combines pre-training on the large, unlabeled CEM500K EM image dataset with a hybrid neural-network model architecture that integrates both local (convolutional layer) and global (transformer layer) image processing to learn from manual image annotations. RETINA outperformed existing models on cellular structure segmentation on five public EM datasets. This improvement works toward automated cellular structure segmentation for the EM community.

## Author summary

Electron microscopy (EM) provides a powerful window into the tiny structures inside cells, but turning these detailed images into meaningful biological insights requires accurately identifying (segmenting) different cellular components. Traditionally, segmentation has been done by hand, a reliable but extremely time-consuming process. To overcome this, we developed RETINA, an innovative method combining advanced deep learning techniques with pre-training on a large collection of EM images called CEM500K. RETINA uniquely integrates two types of neural networks—convolutional

**Funding:** This work was supported by the Canadian Institutes for Health Research (grant PJT 180542 to GDB). The funders had no role in study design, data collection and analysis, decision to publish, or preparation of the manuscript.

**Competing interests:** The authors have declared that no competing interests exist.

networks, which capture detailed local patterns, and transformer networks, which excel at understanding global context. By pre-training on diverse EM images without labels, RETINA learns general features that enable it to quickly adapt and accurately segment new datasets. Our tests across five different EM datasets show that RETINA consistently outperforms existing methods, achieving higher accuracy and faster training. RETINA's robustness and efficiency suggest it could significantly streamline cellular research, helping scientists more rapidly analyze complex microscopic images and uncover new biological discoveries.

## Introduction

Electron microscopy (EM) has revolutionized our understanding of cellular structures at the nanoscale [1]. Three-dimensional volume EM enables detailed visualization of cells, cellular structures and entire organisms [2–5]. Image segmentation (marking pixels/voxels or objects of interest) and annotation (labeling pixels/voxels or objects of interest) is required for EM image analysis and interpretation. Manual segmentation of EM datasets is reliable but labor-intensive [6,7], incentivizing the development of automated segmentation approaches [8]. While many deep learning-based methods have demonstrated expert-level performance for specific segmentation tasks [6,7,9,10], deep learning models often lack generalizable segmentation performance across diverse EM datasets likely due to the wide variability of nanoscale EM structures. Consequently, extensive manual annotation and task-specific fine-tuning remain necessary to adapt existing models to new datasets [11–13], and developing an effective and efficient automated EM segmentation method remains an open challenge.

Supervised deep learning approaches have traditionally been used for image segmentation, where manual identification of pixels/voxels that are part of a given type of object are used to train a model to predict the segmentation of unseen pixels/voxels. This is limited by the availability of manual training data. An improved approach is to use transfer learning to transfer model parameters from similar problems, not necessarily based on EM images. For example, the ImageNet dataset of millions of images [14] can be used to pre-train a model and transfer general image processing ability to new data and tasks [15]. Although ImageNet pre-trained models provide a solid starting point, domain differences in their training data may impact their ability to extract fine-grained features effectively in nanoscale images [16,17]. Research suggests that creating more domain-specific datasets for pre-training can substantially improve performance [18]. Fortunately, open access EM data are increasingly available in central repositories such as EMPIAR [19] and OpenOrganelle [20]. However, most EM-relevant datasets are unlabeled, limiting the use of supervised approaches. As a result, unsupervised pre-training methods, which do not require labels, have been developed and have shown promising results in transfer learning for image analysis, including for EM images [21–23]. For instance, contrastive learning enables the model to associate similar samples and differentiate dissimilar ones by predicting pairs that can be either positive or negative [22,24]. Another effective method to improve image segmentation accuracy is with deep learning model architecture innovations, such as convolutional neural networks (CNN) [25] and transformer architectures [26,27]. CNN-based models, such as U-Net [28] and SegNet [29] excel at extracting local image features [30,31], enabled by hierarchical image feature extraction, shared weights and local receptive fields. Transformers treat an image as a sequence of patches, efficiently capturing global features [27]. Architectures that combine CNN and transformer structures to use both local and global image features can enhance segmentation accuracy. For example, SwinUNETR uses an encoder comprising Swin transformer

blocks combined with a U-Net-like structure capturing hierarchical representation structure, where deeper layers in the encoder correspond to decreased spatial resolution and increased embedding space dimension [32].

To improve segmentation of EM images, we have developed a method that combines the advantages of unsupervised pre-training on large EM-relevant datasets with a hybrid model architecture incorporating both CNN and transformer layers within the encoder. We used the CEM500K dataset, containing $0.5 \times 10^6$ information-rich and heterogeneous EM images [21], for pre-training, and the TransUNet deep learning architecture [33], which features an encoder with integrated CNN and transformer layers, as our model backbone. Given the unlabeled nature of the EM images, a reconstruction-based architecture was established for TransUNet pre-training. Our method was benchmarked against state-of-the-art randomly initialized 2D and 3D deep learning models, as well as a published UNet-ResNet50 model pre-trained on CEM500K by MoCoV2 [34]. All models were evaluated on five public datasets: CREMI Synaptic Clefts [35], Guay [12] (human platelets), Kasthuri++ [36] (neocortex volume), Perez [37] (mammalian brain), and UroCell [38] (urothelium tissue). Our Reconstruction-based prE-trained enhanced TransUNet for electron mIcroscopy segmentatioN on the CEM500K dAtaset (RETINA) model outperforms benchmark models across these diverse datasets.

## Methods

### Model architecture

**RETINA: pre-training.**   To enhance semantic segmentation accuracy in electron microscopy (EM) data, we designed the RETINA model, which pre-trains its encoder on the unlabeled CEM500K dataset [21] and fine-tunes it on benchmark datasets using transferred pre-trained parameters. The RETINA pre-training framework is built upon TransUNet, a model designed for accurate semantic segmentation that uses a hybrid encoder to capture both global and local features [33,39]. Rather than using transformer-based or convolutional layers in isolation, TransUNet first employs convolutional layers to extract high-resolution spatial features from input images, which are subsequently processed by transformer layers to enhance global feature representation. These convolutional feature maps capture fine-grained local details, which are then processed by transformer layers to enhance global feature representation. Unlike standard implementations where patch embeddings are applied directly to the input image, the TransUNet approach applies patch embeddings to the convolutional feature map, enabling the transformer layers to operate on a more informative representation. Within the transformer layers, multi-head self-attention and multi-layer perceptron blocks further enhance feature extraction (Fig 1a). Each patch is mapped to an encoded vector of a predefined dimension, and transitioning from the encoder to the decoder involves reshaping the matrix of patch feature vectors to restore spatial structure. The channel size of the reshaped matrix is then adjusted, followed by progressive upsampling. Skip connections, similar to those in the UNet model, facilitate multi-scale feature aggregation, improving segmentation performance (Fig 1a).

To be efficiently pre-trained on the unlabeled images of CEM500K dataset, RETINA adopts a reconstruction-based self-supervised learning approach. Input 2D images of CEM500K undergo a series of transformations, including flipping, rotation, brightness and contrast adjustments, noise addition, blurring, and patch cropping (S1 Table, Fig 2a). These augmentations can be generally applied to all images and increase image variability and reconstruction difficulty, encouraging the model to learn robust feature representations

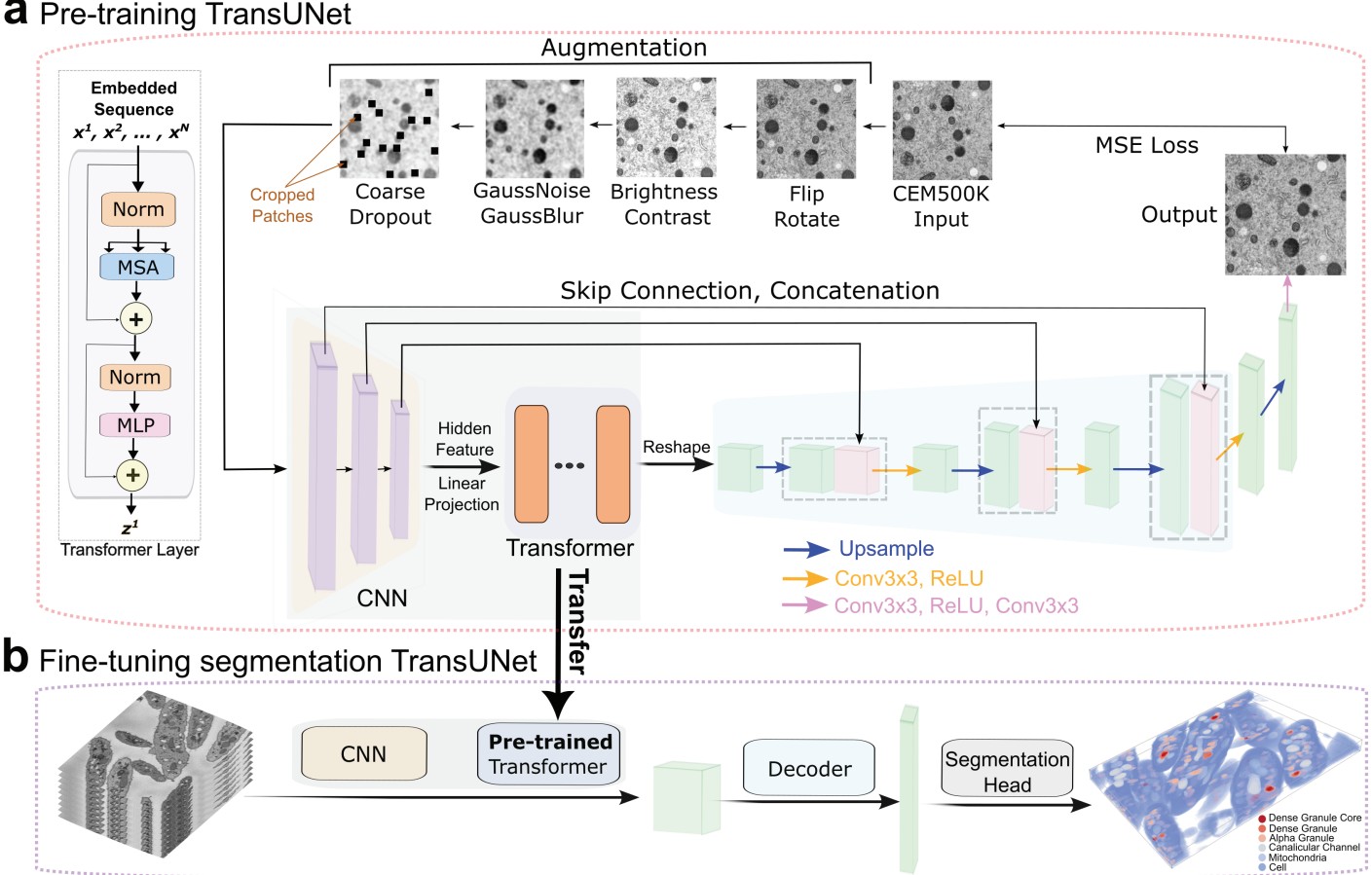

**Fig 1. Overview of the RETINA method. a**, RETINA pre-training process: input images from CEM500K are first augmented and then passed sequentially through convolutional and transformer layers. The embedded features are reshaped and processed by the decoder network, generating reconstructed images. The loss is computed between the input and output images. The structure of the transformer layer is highlighted on the left side. The pre-training process follows this order: augmentation, convolutional layers, transformer layers, reshaping of features, decoding, and loss computation. **b**, RETINA fine-tuning process: after transferring the pre-trained layers, 2D images are encoded by convolutional and transformer layers. The corresponding embedded features are then decoded and labeled via the segmentation head. Abbreviations: CNN, convolutional neural network; Conv, convolution; MLP, multi-layer perceptron; MSA, multi-head self-attention; MSE, mean squared error; Norm, normalization.

across all layers. Since the CEM500K dataset consists of 2D images, we adapted the TransUNet architecture by employing 2D convolutional layers while retaining the original TransUNet encoder-decoder upsampling structure. Skip connections are preserved to enhance feature aggregation across different resolution levels. Instead of using the standard TransUNet segmentation head, it is replaced with a convolutional decoder that reconstructs the input image, aligning with the reconstruction-based pre-training task. The reconstructed output is compared to the original image using mean squared error after applying the same geometric transformations used for augmentation. The complete set of RETINA pre-training parameters is provided in S2 Table.

This pre-training strategy on 2D images ensures compatibility with the fine-tuning phase, enabling seamless transfer of learned representations due to the shared encoder structure. By training on the CEM500K image dataset, RETINA learns to reconstruct altered images, facilitating a deeper understanding of structural details and global context by distinguishing

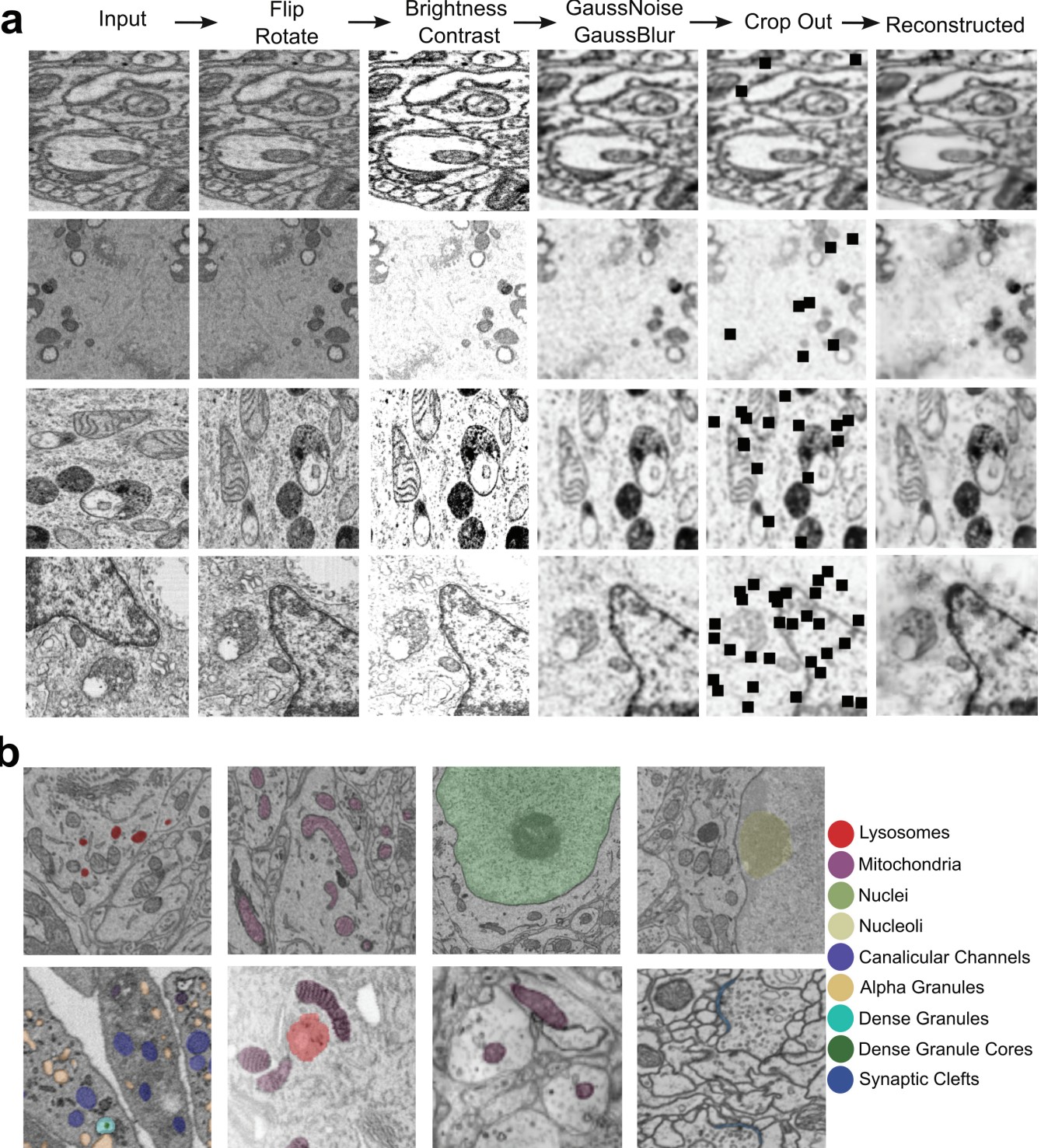

**Fig 2. Augmented CEM500K images and benchmark datasets.** **a**, Example images of CEM500K and their augmented counterparts for each augmentation step: from left to right, the columns show the Input, flip and rotate, brightness and contrast adjustment, Gaussian noise and Gaussian blur, cropping, and final reconstructed images. Each row represents an example image undergoing the entire augmentation process. **b**, Example images and label maps of each of the five benchmark datasets: the first row represents lysosomes, mitochondria, nuclei, and nucleoli in the Perez dataset. The second row corresponds to the Guay, UroCell, Kasthuri++, and CREMI Synaptic Clefts datasets.

between original and altered images. Further details on the model architecture and implementation are provided in S1 Text.

**Fine-tuning using TransUNet.** To use the benefits of pre-training on the CEM500K dataset, transfer learning is applied. Since the pre-trained layers are 2D-based, the fine-tuning architecture must also be compatible with 2D inputs. Therefore, the 2D TransUNet is employed for RETINA fine-tuning. For 3D input images, the volume is first sliced into 2D images before being processed by the model. The pre-trained transformer layer parameters are then transferred to the corresponding transformer layers (Fig 1b). To ensure stability during training on benchmark tasks, these transferred parameters are frozen (not updated further). Since the fine-tuning task involves semantic segmentation, consistent with TransUNet [33,39], and requires pixel-wise mask generation, a segmentation head is used as the final layer to assign each pixel a class ID corresponding to its predicted category. This adaptation of TransUNet, from reconstruction to segmentation, ensures that the model accurately performs the segmentation task while using the pre-trained parameters. The RETINA implementation parameters for fine-tuning are listed in S3 Table. Details about the model implementation can be found in S1 Text.

## Dataset

**CEM500K used for pre-training.** To improve the accuracy of EM image segmentation, we used the CEM500K dataset for model pre-training. It is an information-rich and non-redundant 25 gigabyte 2D EM image dataset [21]. Briefly, $5.3 \times 10^6$ images were collected and cropped into $224 \times 224$ pixel patches, forming the CEMraw dataset. This dataset contains a wide range of modalities and sample preparation protocols (a complete list of the datasets with related attribution can be found in the supplementary materials of the CEM500K paper [21]). The images underwent deduplication and filtering, resulting in the CEMdedup and final CEM500K datasets. The selected $0.5 \times 10^6$ images that we use for training contain diverse cellular images from more than 100 unrelated biological projects [21], as shown by representative images (Fig 2a, left column).

All $0.5 \times 10^6$ images in the CEM500K dataset were used as input for RETINA pre-training and processed with augmentation procedures including flipping, rotation, brightness and contrast adjustment, noise addition, blurring, and patch cropping. Flipping and rotation were randomly applied with the probability 0.5. In addition, each operation has a set of parameters (S1 Table) to define the degree of augmentation. Each image went through different degrees of augmentation controlled by these parameters. This approach increases randomization to enhance data variability. For example, as shown in (Fig 2a), the first image is neither flipped nor rotated; the second is only flipped; the third is only rotated; and the fourth is both flipped and rotated. Since noise and blur were added after brightness adjustment, the images appear less bright in the fourth column. As shown in the fifth column, the number of cropped patches is randomly applied in the range from 1 to 32. For better visualization, the four representative CEM500K images input to the pre-training process are sorted according to the number of cropped patches, from the smallest to the largest (four rows in Fig 2a).

**Five benchmark datasets used for fine-tuning.** We evaluated RETINA's performance using five benchmark datasets: CREMI Synaptic Clefts [35], Guay [12] (human platelets), Kasthuri++ [36] (neocortex volume), Perez [37] (mammalian brain), and UroCell [38] (urothelium tissue) (S4 Table). These datasets include nine subcellular structures for segmentation: lysosomes, mitochondria, nuclei, nucleoli, canalicular channels, alpha granules, dense granules, dense granule cores, and synaptic clefts. Representative images and ground truth labels for each dataset are presented in Fig 2b, with labels overlaid on the corresponding raw

images. Guay and UroCell have multiple organelles labeled for segmentation. To ensure consistency with previous research [21], volumes A and B were used for training, while volume C was used for testing, as the test data for the CREMI Synaptic Cleft benchmark is not publicly available. Meanwhile, the four label types in the Perez dataset were treated as independent binary segmentation tasks [21]. Additionally, these datasets use different electron microscopy imaging technologies (S4 Table), increasing the diversity of benchmarking data.

## Results

### Evaluation of RETINA on benchmark data

**RETINA achieves the best segmentation accuracy on EM benchmarks.** To evaluate the segmentation performance of RETINA, it was benchmarked against several state-of-the-art models for EM image segmentation. The UNet-ResNet50 model [40], pre-trained on CEM500K using MoCoV2 [23], was selected for its superior performance over both the randomly initialized version and the version pre-trained with supervised learning on ImageNet [21]. To ensure parity in training, RETINA underwent 200 pre-training epochs, consistent with the UNet-ResNet50 pre-trained on CEM500K model [21]. Additionally, to highlight the benefits of pre-training and RETINA's hybrid encoder architecture, a randomly initialized 2D TransUNet model was included as a benchmark. The randomly initialized UNet-ResNet50 model serves as the baseline, enabling a systematic assessment of RETINA's improvements. Specifically, comparing the baseline to the UNet-ResNet50 pre-trained on CEM500K model quantifies the impact of pre-training, while comparing the baseline to TransUNet isolates the advantage of RETINA's encoder structure. The overall enhancement of RETINA is demonstrated by comparing it against all other models. To further assess RETINA's ability to handle 3D images, two high-performing 3D models are included: 3D TransUNet [39] and nnUNet with the configuration of 3D full resolution [41,42]. The number of training iterations (each iteration representing a batch-level update with a corresponding learning rate adjustment) was set to ensure sufficient fine-tuning. This allowed the models to converge effectively and achieve stable Intersection over Union (IoU) scores (predicted segmentation vs. ground truth segmentation at the pixel level). For consistency and comparison purposes, we selected the iteration counts based on previous work [21]. The corresponding training iteration numbers are listed in Table 1. We reimplemented each benchmark model using the same code and parameters as published [21]. Although the Perez dataset includes several labeled organelles, each was trained independently as a binary task and averaged to obtain the final IoU value, consistent with prior work [21]. Table 1 presents the IoU values for all benchmark datasets. The reported metrics correspond to inference results obtained using checkpoints from the specified number of training iterations. The evaluation is also presented based on other semantic metrics, including F-score (S5 Table), Precision (S6 Table), Recall (S7 Table), and Mean False Distance (MFD) (S8 Table). The description of these metrics can be found in S1 Text. For isotropic 3D segmentation in the UroCell dataset, the model was trained on a combination of axial (xy), sagittal (yz), and coronal (xz) slices. During inference, orthoplane prediction is applied [21], where the model generates predictions on the *xy*, *xz*, and *yz* planes, and the outputs are averaged to integrate 3D information. For anisotropic volumes, including the CREMI Synaptic Clefts, Guay, and Kasthuri++, predictions are made only on *xy* cross-sections. In the Perez dataset, since images were randomly sampled in 2D [37], benchmarking with 3D-based models is excluded.

A high IoU score indicates greater overlap between the predicted segmentation and the ground truth. For the CREMI Synaptic Clefts dataset, pre-training UNet-ResNet50 on CEM500K improves performance compared to the randomly initialized UNet-ResNet50

**Table 1. Comparison of segmentation IoU values for RETINA versus benchmark models, including: randomly initialized (Rand. Init.) UNet-ResNet50, UNet-ResNet50 pre-trained on CEM500K, Rand. Init. 2D TransUNet, Rand. Init. 3D TransUNet and Rand. Init. nnUNet.** They are the inference results based on the checkpoints of their corresponding training iterations. Lysosomes, mitochondria, nuclei, and nucleoli within Perez benchmark are listed separately. Mean values of three independent runs are reported.

| Benchmark | Training Iterations | Rand. Init. UNet-ResNet50 | CEM500K UNet-ResNet50 | Rand. Init. 2D TransUNet | Rand. Init. 3D TransUNet | Rand. Init. nnUNet | RETINA |
|---|---|---|---|---|---|---|---|
| CREMI S.C. | 5000 | 0.000 | 0.246 | 0.294 | 0.313 | 0.243 | **0.327** |
| Guay | 2500 | 0.452 | 0.468 | 0.266 | 0.493 | 0.495 | **0.552** |
| Kasthuri++ | 10000 | 0.904 | 0.915 | 0.894 | 0.905 | 0.898 | **0.916** |
| Perez | 2500 | 0.850 | 0.905 | 0.893 | – | – | **0.919** |
| Lysosomes | – | 0.845 | 0.854 | 0.845 | – | – | **0.885** |
| Mitochondria | – | 0.841 | 0.887 | 0.847 | – | – | **0.890** |
| Nuclei | – | 0.984 | 0.990 | 0.990 | – | – | **0.991** |
| Nucleoli | – | 0.731 | 0.889 | 0.891 | – | – | **0.910** |
| UroCell | 1000 | 0.203 | 0.598 | 0.579 | 0.573 | 0.567 | **0.610** |

model, increasing the IoU from 0.000 to 0.246. However, despite this improvement, the pre-trained UNet-ResNet50 still underperforms relative to the randomly initialized 2D TransUNet, highlighting the superior feature extraction capabilities of the hybrid model structure. Although both 3D TransUNet and nnUNet are 3D-based models, their performance remains inferior to RETINA. The same metrics as the CREMI challenge are also used (S9 Table), and RETINA also has the best performance among all models. On the Guay dataset, the randomly initialized 2D TransUNet performs worse than the randomly initialized UNet-ResNet50 baseline. However, with pre-training, RETINA achieves the highest IoU among all benchmark models, including the two 3D models, demonstrating the effectiveness of pre-training. For the Perez and Kasthuri++ benchmarks, RETINA continues to outperform other models, though the improvements are less pronounced, likely due to the relative simplicity of these binary segmentation tasks. The UroCell benchmark shows that RETINA has the IoU 0.610 which is the highest compared to other models. The impact of pre-training on CEM500K is evident, as the pre-trained UNet-ResNet50 achieves an IoU of 0.598, outperforming both the randomly initialized 2D and 3D models. Other evaluation metrics, including F-score (S5 Table), Precision (S6 Table), Recall (S7 Table), and MFD (S8 Table), follow similar performance trends across models. Overall, these results show that RETINA consistently achieves the best segmentation performance among the benchmark models.

**RETINA converges quickly.** In many application areas, models can be found to achieve strong performance with a few fine-tuning steps if they are pre-trained [15]. We tested if this holds with RETINA by evaluating the performance (IoU) changes of all tested models and datasets at different numbers of fine-tuning iterations (Fig 3a). In terms of method comparison using pre-training with different model architectures, we evaluated RETINA against the UNet-ResNet50 pre-trained on CEM500K across five benchmark datasets. On the CREMI Synaptic Clefts dataset, RETINA achieves an IoU of 0.330 within 1000 iterations, maintaining high performance with minimal additional gains in subsequent iterations (Fig 3a). In contrast, the UNet-ResNet50 pre-trained on CEM500K reaches comparable results by 3000 iterations. On the Kasthuri++ dataset, both pre-trained models stabilize by 400 iterations, but RETINA reaches a higher IoU value earlier. For the simpler binary segmentation of the Perez dataset, RETINA reaches a high performing state early on, while the UNet-ResNet50 pre-trained on CEM500K converges rapidly but lags 100 iterations behind RETINA (Fig 3a). On the UroCell dataset, RETINA attains an IoU of over 0.6 within 200 iterations, whereas the UNet-ResNet50 pre-trained on CEM500K is slower to converge, improving from 400 to

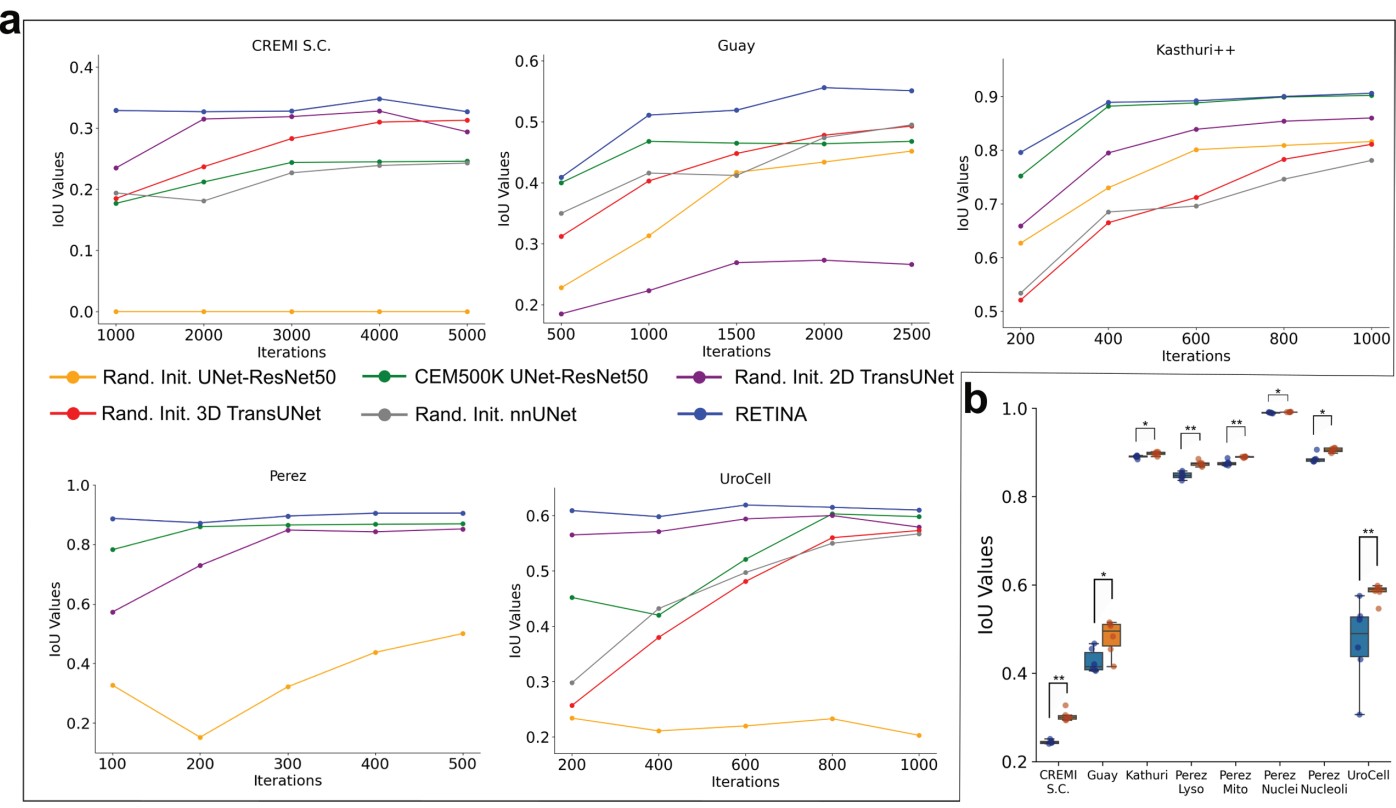

**Fig 3. RETINA demonstrates fast convergence and robustness, outperforming other models across all segmentation benchmarks.** **a**, IoU values of RETINA and other benchmark models across different fine-tuning iterations. Each IoU value is obtained by inferring on test datasets using model weights trained with the specified number of iterations. For better visualization, the iteration range is chosen to capture the model progression toward convergence in each dataset. **b**, Box plots representing the distribution of IoU values for each benchmark, with each benchmark consisting of six experiments corresponding to different random seeds. The UNet-ResNet50 pre-trained on CEM500K model is colored in blue, and RETINA is in orange. *p<0.05, **p<0.01.

800 iterations (Fig 3a). Overall, RETINA converges more rapidly than the UNet-ResNet50 pre-trained on CEM500K in most experiments.

To evaluate RETINA against randomly initialized benchmark models, its performance was compared to randomly initialized 2D TransUNet, UNet-ResNet50, 3D TransUNet, and nnUNet across five datasets. Since the Perez dataset consists of randomly sampled 2D images without 3D structural information, the two 3D benchmark models were not applied to this dataset. On the CREMI Synaptic Clefts dataset, the randomly initialized 2D TransUNet requires 2000 iterations to reach peak performance, 1000 iterations slower than RETINA. The randomly initialized UNet-ResNet50 baseline consistently produces an IoU of 0.000 (Fig 3a), as previously reported [21], underscoring the dataset's complexity and making it unsuitable for convergence speed comparisons. The two 3D models exhibit a gradual increase in performance, approaching convergence at approximately 5000 iterations. On the Guay dataset, RETINA stabilizes around 2000 iterations, while the randomly initialized UNet-ResNet50 model converges more slowly, reaching stability at approximately 2500 iterations. TransUNet stabilizes earlier at around 1500 iterations but with significantly lower accuracy, making direct convergence comparisons less informative (Fig 3a). For the Kasthuri++ dataset, randomly initialized 2D models continue improving beyond 600 iterations, whereas RETINA reaches

stability earlier. However, the randomly initialized 3D models do not show clear signs of convergence and continue increasing in IoU beyond 1000 iterations. On the Perez dataset, both randomly initialized 2D models require more iterations to converge compared to RETINA. Similarly, on the UroCell dataset, the randomly initialized 2D TransUNet exhibits gradual improvement up to 600 iterations (Fig 3a), though with a shallow convergence slope. The UNet-ResNet50 baseline performs poorly across the 200–1000 iteration range, making meaningful comparisons on this dataset less relevant. The 3D models show a slower performance increase after 800 iterations compared to their steeper improvement in earlier stages. Across most experiments, RETINA converges faster than all randomly initialized models. In summary, RETINA consistently achieves high performance more quickly across five benchmark datasets compared to all other models tested.

**RETINA is robust to random seed changes.** RETINA exhibits superior segmentation accuracy on EM datasets by leveraging pre-training on CEM500K and using the TransUNet backbone. The initial experiments, however, were conducted using a single random seed across all models. To assess the reproducibility of our results, we repeated the fine-tuning of RETINA and the MoCoV2 pre-trained benchmark model on each dataset using six different random seeds. The resulting IoU score distributions demonstrate that RETINA consistently significantly outperforms the UNet-ResNet50 pre-trained on CEM500K across all datasets (Mann-Whitney U test, Fig 3b). Notably, RETINA exhibits substantially lower variance in the UroCell multiclass benchmark, underscoring its ability to reduce uncertainty in more complex multi-class segmentation tasks. Across the other benchmarks, RETINA maintains consistently low variance across different random seeds. Overall, RETINA significantly outperforms the UNet-ResNet50 pre-trained on CEM500K in accuracy and is robust to random seed variation.

**RETINA evaluation by manual image inspection.** To visually assess segmentation quality, we manually inspected representative label maps for each benchmark. We reviewed images in two and three dimensions (2D and 3D). For 2D visualization, representative predicted images from the Perez, Kasthuri++, and UroCell datasets are shown in Fig 4a. For 3D visualization, the Guay and CREMI Synaptic Clefts datasets highlight nanoscale labeled structures: Guay includes seven distinct organelles, while CREMI features synaptic cleft labels (Fig 4b). 3D visualizations of Kasthuri++ and UroCell are provided in S1 Fig. All visualizations were generated from the models used in the RETINA benchmarking experiments as detailed in Table 1.

For the 2D inferred visualization, we manually identified improved segmentation regions among the models, demonstrating that RETINA produces output masks closest to the ground truth in the first row (Fig 4a, dashed-outline squares). For example, comparing with the ground truth images (top row), RETINA successfully identifies lysosomes highlighted in cyan in the last row of Fig 4a; however, the TransUNet (second row) and UNet-ResNet50 pre-trained on CEM500K (third row) are not able to detect the organelles in these highlighted regions. While nucleoli segmentation in the Perez dataset shows no significant differences among the models, RETINA excels in the relatively challenging regions of the Kasthuri++ dataset, where both TransUNet and the UNet-ResNet50 pre-trained on CEM500K fail (yellow squares). RETINA also accurately identifies the lysosome region marked with blue squares, whereas TransUnet identifies less area and the UNet-ResNet50 pre-trained on CEM500K misassigns the region as mitochondria (purple) instead of lysosomes (red). RETINA's ability to more thoroughly identify entire target objects and avoid mislabeling helps explain its higher IoU values compared to other benchmark models (Table 1).

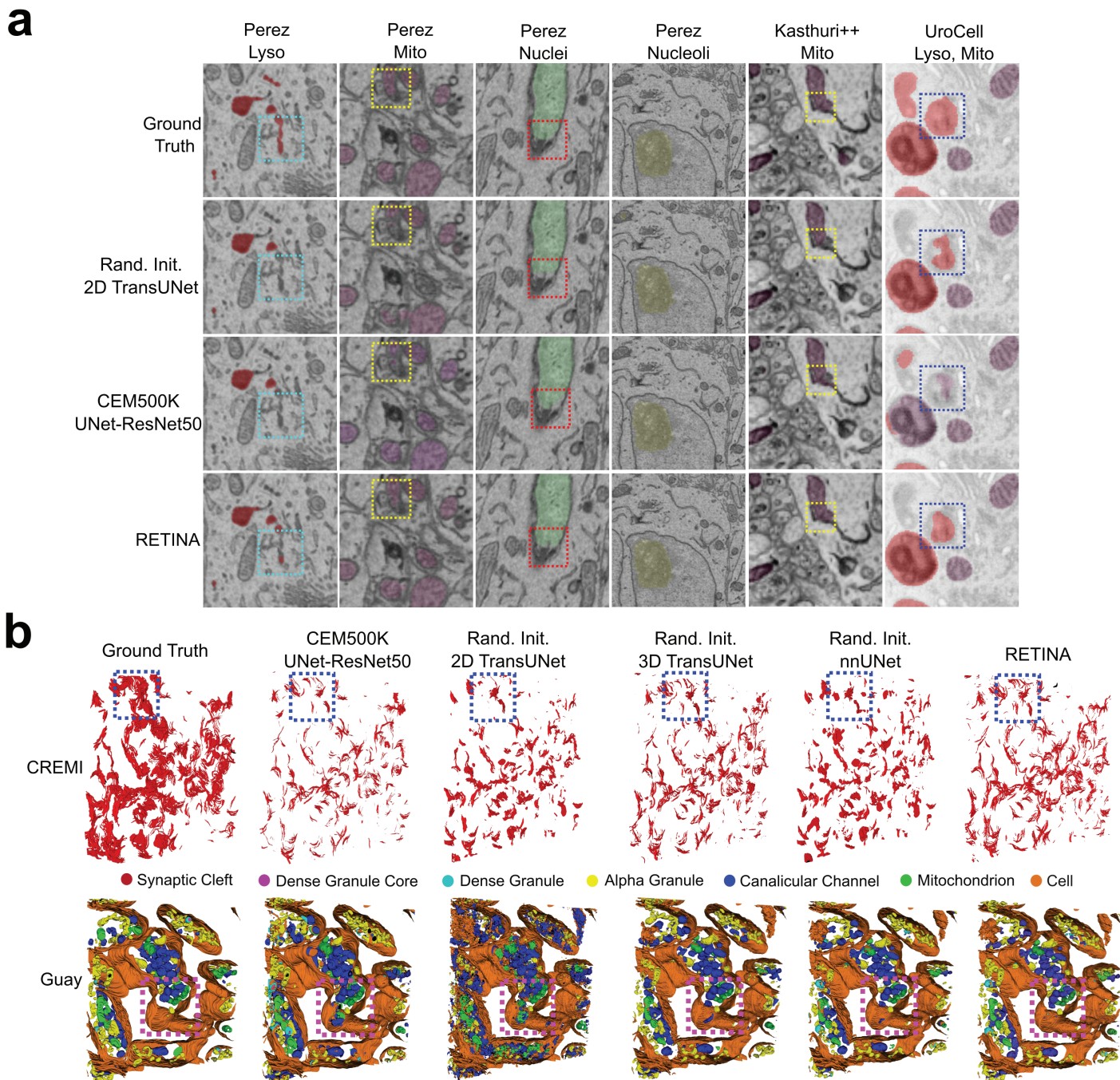

**Fig 4. Visual comparison of segmentation. a**, Example segmentations from all 2D benchmark datasets. The first row shows the ground truth; the second row represents the randomly initialized TransUNet; the third row depicts the UNet-ResNet50 pretrained on CEM500K model; the fourth row corresponds to RETINA. Lysosomes are shown in red, mitochondria in purple, nuclei in green, and nucleoli in yellow. **b**, Example segmentations shown in 3D view. Labels for the CREMI Synaptic Clefts and Guay datasets are shown in the first and second rows, respectively. The dashed-line squares highlight regions where RETINA demonstrates improved segmentation accuracy. Abbreviations: Lyso, Lysosomes; Mito, Mitochondria.

For 3D visualization of the CREMI Synaptic Clefts dataset (top row, Fig 4b), RETINA demonstrates superior segmentation accuracy across representative regions, such as those highlighted by blue squares. Compared to other models, RETINA yields denser and more anatomically precise synaptic cleft labeling, showing closer alignment with the ground truth (left column, Fig 4b). While the randomly initialized 3D TransUNet exhibits notable improvement, RETINA achieves the most accurate and detailed segmentation results among all benchmark models. In the Guay dataset, the regions highlighted by purple squares reveal that all benchmark models produce numerous false labels (second to fifth columns in Fig 4b). Specifically, the ground truth in these areas primarily represents pure cell regions (uncolored) without additional labeled cellular structures. However, the models erroneously label a number of canalicular channels (blue) and mitochondria (light green) within these regions (Fig 4b), though the two 3D models exhibit fewer mislabelings. In contrast, RETINA (last column in Fig 4b) demonstrates minimal mislabeling, closely aligning with the ground truth. In conclusion, both 2D and 3D segmentation visualizations demonstrate that RETINA effectively captures organelle features.

## RETINA performs well benefiting from the CEM500K dataset

One of the benefits of RETINA is its ability to pre-learn features from the CEM500K dataset. To further explore the importance of pre-training on CEM500K, we experimented with different pre-training dataset selections, including ImageNet, CEM500K, and a combination of both (S10 Table), while keeping other parameters constant. Given that ImageNet is a labeled dataset, pre-training was supervised and applied to the TransUNet model. For the model pre-trained on both ImageNet and CEM500K, it was first pre-trained on ImageNet, and then the parameters were transferred to the reconstruction-based architecture (Fig 1a) for further pre-training on the CEM500K dataset. Pre-training on the CEM500K dataset results in better performance, as indicated by higher IoU scores on the CREMI Synaptic Clefts, Guay, Kasthuri++, and Perez datasets, whereas ImageNet pre-training is superior for the Uro-Cell dataset (S10 Table). Our findings show that CEM500K pre-training generally outperforms ImageNet pre-training, with the exception of the UroCell dataset, where ImageNet pre-training achieves superior results. The diminished performance of the CEM500K pre-trained model on UroCell suggests that the domain-specific advantages of CEM500K are less applicable to this dataset compared with the other benchmark datasets. Additionally, a combined pre-training approach using both ImageNet and CEM500K does not surpass CEM500K pre-training alone across the benchmark datasets (S10 Table), with only a slight improvement found for mitochondrial segmentation in the Perez benchmark. Thus, combined pre-training does not offer substantial benefits, CEM500K pre-training remains a robust strategy for enhancing segmentation accuracy across most datasets.

## Discussion

RETINA enhances the accuracy of segmentation on EM-related cellular images, benefiting from two key aspects. First, RETINA uses a TransUNet backbone, with its hybrid encoder (comprising convolutional and transformer layers) achieving both local and global image feature extraction. Second, it is pre-trained on the CEM500K dataset, a diverse and non-redundant EM image dataset, that helps RETINA to pre-learn EM-related features and infer the nanoscale objects accurately after fine-tuning on the benchmarks.

Given that CEM500K is an unlabeled dataset, an unsupervised pre-training design is required. Initially, instead of the reconstruction-based unsupervised pre-training method applied for RETINA, we employed SimCLR due to its proven effectiveness in pre-training

convolutional layers [22]. After pre-training, we transferred the pre-trained convolutional layers within the encoder to the fine-tuning phase. However, this approach did not yield any improvement over the randomly initialized TransUNet. Similarly, using masked autoencoders [43] to pre-train only the transformer layers resulted in poor outcomes. These findings suggest that pre-training only parts of the hybrid encoder is ineffective; instead, the entire encoder should be pre-trained. The complexity of the hybrid encoder makes it challenging to achieve good segmentation performance on EM images with pre-training, as the hybrid structure requires integration, consistency, and transition within the encoder [44]. In response, we developed a reconstruction-based pre-training method that recovers augmented images to their original state, involving both convolutional and transformer layers, thereby enabling consistent pre-training of the hybrid model. This reconstruction-based method should be generalizable to other hybrid models, such as U-Mamba [45], for pre-training on unlabeled datasets.

The similar model structure between the RETINA pre-training and fine-tuning phases should enable seamless transferring of the pre-trained layers from the corresponding pre-training phase to the fine-tuning phase. However, transferring all parameters did not result in improved performance; instead, the best results were achieved by transferring only the transformer layer parameters. This may be because more adjustable parameters are required to accommodate the information learned during pre-training and fine-tuning. In the RETINA design, the convolutional layers remain unfrozen during fine-tuning, allowing them to adapt to the fine-tuning dataset, while only the transformer layers are frozen. This approach enables the convolutional layers to adjust and learn new features during fine-tuning, making RETINA more flexible and adaptable. At the same time, the features learned from CEM500K during pre-training are preserved by freezing the transformer layers, preventing them from being altered during fine-tuning. This strategy appears to balance the retention of pre-trained features with the adaptation to new data, resulting in more accurate segmentation.

Given that augmentation can increase the diversity of the dataset and modify the difficulty of the training task, it needs to be carefully designed [46]. The set of augmentation operations can be customized based on different pre-training datasets and downstream tasks to maximize the benefits obtained from pre-training. For RETINA, we employed random flips, rotations, brightness, and contrast adjustments as standard methods to augment the dataset. Additionally, adding noise and blur increased the reconstruction difficulty. Inspired by the masked autoencoder model [43], we also randomly masked parts of the input images to enhance feature learning. Effective augmentation ensures that RETINA's deeper layers are thoroughly trained. Given the skip connection design of TransUNet, insufficiently robust augmentation may cause the input image vectors to bypass the deep layers, leading to quick convergence without adequately updating the deeper layers (Fig 1a), resulting in insufficient pre-training and no MSE loss improvement.

One important benefit of pre-training is that it can reduce the time required for training downstream tasks [15]. We show that RETINA converges faster than benchmark models without pre-training (Fig 3a). Since times per iteration are nearly identical across models (S11 Table), RETINA's faster convergence indicates that it requires fewer iterations to achieve high performance, reducing overall training time. In the CREMI Synaptic Clefts, Perez, and UroCell benchmarks, RETINA demonstrates rapid convergence within just a few hundred iterations. The required number of fine-tuning iterations may vary depending on the complexity of the tasks and the relevance of features between the pre-training and fine-tuning datasets. In our experimental design, we set the number of fine-tuning iterations based on

prior work [21] to ensure a consistent basis for performance comparisons. For future applications of RETINA, fewer fine-tuning iterations could be used to reduce runtime, making the model more efficient.

The choice of pre-training dataset influences the final segmentation accuracy. Although the CEM500K dataset contains $0.5 \times 10^6$ diverse and informative cellular EM images, there remains potential to develop more powerful pre-training datasets with broader generalizability, such as the recently published CEM1.5M dataset [47]. In addition, when the downstream task involves more images similar to the pre-training images, the model can more efficiently capture relevant features, potentially outperforming models pre-trained on large but unrelated datasets [48]. The image similarity between the pre-training dataset and the fine-tuning dataset can be influenced not only by the type of target objects they share but also by their optical properties, such as brightness, contrast, and white balance [49]. This may explain why pre-training on CEM500K is less effective for the UroCell dataset compared to supervised pre-training on large ImageNet datasets (S10 Table), as UroCell images have higher brightness compared to all other benchmarks Figs 2b and 4a). Therefore, there is potential to create more suitable pre-training datasets that balance generalization and task-specific features based on downstream tasks. Nevertheless, RETINA's hybrid encoder structure helps mitigate the limitations of pre-training datasets. For instance, in the CREMI Synaptic Clefts benchmark, the MoCoV2 pre-trained UNet-ResNet50 model improved the IoU score from 0.000 to 0.246 (Table 1), while RETINA, without any pre-training on CEM500K, achieved an even higher IoU score of 0.294. This suggests that the hybrid encoder structure can be more important than pre-training, especially when the features of the pre-training dataset are not well-aligned with those of the fine-tuning dataset, and the benefits of pre-training may be diminished in such cases.

Although RETINA outperforms benchmark 3D models, its performance could be further enhanced by incorporating 3D structures and pre-training on a fully 3D dataset. Currently, CEM500K consists solely of 2D images, limiting RETINA's ability to capture 3D EM features along the z-axis from its pretraining set. However, with the rapid expansion of EM datasets, the emergence of large-scale volume EM datasets, such as those available on OpenOrganelle [50], has broadened the scope of 3D EM data. Well-curated huge 3D EM datasets could be generated to facilitate pre-training on volumetric data, enabling more comprehensive feature learning. In this scenario, RETINA could be extended with a fully 3D model architecture to further improve its performance. Furthermore, achieving generalizable performance often requires large-scale pre-training datasets, making manual mask curation impractical. As a result, developing effective pre-training methods that do not rely on labeled data, such as the approach proposed here, will be essential for advancing self-supervised learning in EM segmentation.

Since RETINA is primarily designed for semantic segmentation, being built upon the TransUNet architecture, we focus exclusively on this task, where each pixel or voxel is assigned a class identifier (ID). Accordingly, we only considered semantic segmentation benchmarks and evaluation metrics. Applying RETINA to instance segmentation, which requires distinguishing individual objects with unique IDs, would necessitate additional post-processing steps (e.g., the Watershed algorithm [51]) and extensive hyperparameter tuning which was not done in all of our comparator models. Thus, we limited our experiments to end-to-end semantic segmentation tasks, emphasizing the effects of pre-training and architectural design. Nonetheless, RETINA can be adapted for instance segmentation with appropriate post-processing applied to its semantic predictions. Furthermore, the RETINA pre-training method could be extended to support the pre-training of the end-to-end instance segmentation models such as Mask R-CNN [52].

We aim to develop a method that enhances EM image segmentation using a hybrid model architecture and pre-training. While our reconstruction-based approach demonstrates strong performance using the TransUNet backbone, future directions could explore applying this strategy to other hybrid encoder architectures, such as U-Mamba [45] and SwinUNETR [32], to potentially achieve further improvements. Additionally, this pre-training framework could be extended to models designed for other tasks, including instance segmentation. Moreover, the RETINA pre-training and fine-tuning pipeline is applicable to a wide range of EM datasets, both existing and those that may emerge in the future, offering a promising approach to performance enhancement. The released RETINA pre-trained weights also serve as a resource for the EM research community, facilitating more effective and efficient segmentation of EM images.

## Supporting information

**S1 Fig. 3D visual comparison of segmentation.** Labels for the Kasthuri++ and UroCell datasets are shown in the first and second rows, respectively. The dashed-line squares highlight regions where RETINA demonstrates improved segmentation accuracy.
(TIFF)

**S1 Text. Description of RETINA pre-training methods, RETINA fine-tuning methods, inference methods, and benchmark model implementation.**
(PDF)

**S1 Table. RETINA augmentation function settings.** The first column lists the function name. The second column shows the probability of each function being applied to the image. The third column indicates the augmentation set number to which each function belongs. Set 1 comprises the augmentation operations applied to both input images, ImageA and ImageB, as outlined in the RETINA pre-training methods section. Set 2 includes the operations applied exclusively to ImageA. The last column details the specific parameter settings for each function.
(PDF)

**S2 Table. RETINA implementation parameters for pre-training.**
(PDF)

**S3 Table. RETINA implementation parameters for fine-tuning.**
(PDF)

**S4 Table. Characteristics of the benchmark datasets.** Abbreviations: ssTEM, serial section transmission electron microscopy; SBFSEM, serial block-face scanning electron microscopy; AT-SEM, array tomography scanning electron microscopy; FIBSEM, focused ion beam scanning electron microscopy.
(PDF)

**S5 Table. Comparison of segmentation F-score for RETINA versus benchmark models, including: randomly initialized (Rand. Init.) UNet-ResNet50, UNet-ResNet50 pre-trained on CEM500K, Rand. Init. 2D TransUNet, Rand. Init. 3D TransUNet and Rand. Init. nnUNet.** Lysosomes, mitochondria, nuclei, and nucleoli within Perez benchmark are listed separately. Mean values of three independent runs are reported.
(PDF)

**S6 Table. Comparison of segmentation Precision values for RETINA versus benchmark models, including: randomly initialized (Rand. Init.) UNet-ResNet50, UNet-ResNet50**

**pre-trained on CEM500K, Rand. Init. 2D TransUNet, Rand. Init. 3D TransUNet and Rand. Init. nnUNet.** Lysosomes, mitochondria, nuclei, and nucleoli within Perez benchmark are listed separately. Mean values of three independent runs are reported.
(PDF)

**S7 Table. Comparison of segmentation Recall values for RETINA versus benchmark models, including: randomly initialized (Rand. Init.) UNet-ResNet50, UNet-ResNet50 pre-trained on CEM500K, Rand. Init. 2D TransUNet, Rand. Init. 3D TransUNet and Rand. Init. nnUNet.** Lysosomes, mitochondria, nuclei, and nucleoli within Perez benchmark are listed separately. Mean values of three independent runs are reported.
(PDF)

**S8 Table. Comparison of segmentation Mean False Distance in pixel/voxel distance for RETINA versus benchmark models, including: randomly initialized (Rand. Init.) UNet-ResNet50, UNet-ResNet50 pre-trained on CEM500K, Rand. Init. 2D TransUNet, Rand. Init. 3D TransUNet and Rand. Init. nnUNet.** The Mean False Distance is based on the mean of the bidirectional directed Hausdorff Distance between the ground truth and the predicted segmentation. Lysosomes, mitochondria, nuclei, and nucleoli within Perez benchmark are listed separately. Mean values of three independent runs are reported.
(PDF)

**S9 Table. Performance on the CREMI dataset using the same evaluation metrics as the challenge.** Metrics include the CREMI score (mean of ADGT and ADF), false positives (FP), false negatives (FN), 1 - F-score, ADGT (average distance of any predicted cleft voxel to the closest ground truth cleft voxel), and ADF (average distance of any ground truth cleft voxel to the closest predicted cleft voxel). Lower values indicate better performance. Models were trained on volumes A and B and evaluated on volume C. The performance of the randomly initialized UNet-ResNet50 could not be calculated due to its low accuracy.
(PDF)

**S10 Table. Comparison of IoU scores for models pre-trained on different datasets.** The models include randomly initialized (Rand. Init.) 2D TransUNet, 2D TransUNet pre-trained on ImageNet, 2D TransUNet using ImageNet pre-trained parameters followed by pre-training on CEM500K, and RETINA. These models were fine-tuned and evaluated on all benchmark datasets listed in the first column. The IoU scores represent the best performance achieved with the specific number of training iterations shown in the second column. Lysosomes, mitochondria, nuclei, and nucleoli within Perez benchmark are listed separately. Mean values of three independent runs are reported.
(PDF)

**S11 Table. Average training time per iteration for each model on each dataset, measured using Nvidia A100 GPUs on the Narval cluster of the Digital Research Alliance of Canada.** The values represent the mean time per iteration, averaged over the entire training process. Rand. Init., randomly initialized.
(PDF)

## Acknowledgments

All authors thank the Digital Research Alliance of Canada for high-performance computing access.

## Author contributions

**Conceptualization:** Cheng Xing, Ronald Xie.

**Data curation:** Cheng Xing.

**Formal analysis:** Cheng Xing.

**Funding acquisition:** Gary D. Bader.

**Investigation:** Cheng Xing.

**Methodology:** Cheng Xing.

**Project administration:** Gary D. Bader.

**Resources:** Gary D. Bader.

**Supervision:** Ronald Xie, Gary D. Bader.

**Validation:** Cheng Xing.

**Visualization:** Cheng Xing.

**Writing – original draft:** Cheng Xing.

**Writing – review & editing:** Cheng Xing, Gary D. Bader.

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
