## [Decision Letter · Decision Letter 0]

12 Feb 2025

PCOMPBIOL-D-24-01938

RETINA: Reconstruction-based Pre-Trained Enhanced TransUNet for Electron Microscopy Segmentation on the CEM500K Dataset

PLOS Computational Biology

Dear Dr. Xing,

Thank you for submitting your manuscript to PLOS Computational Biology. After careful consideration, we feel that it has merit but does not fully meet PLOS Computational Biology's publication criteria as it currently stands. Therefore, we invite you to submit a revised version of the manuscript that addresses the points raised during the review process.

Please note that both reviewers expressed enthusiasm for hte topic, but both had signfiicant reservations about the rigor of the work.  A revision would need to do much to address the concerns of the reviewers, but the work to address each point does seem feasible.

Please submit your revised manuscript within 60 days Apr 14 2025 11:59PM. If you will need more time than this to complete your revisions, please reply to this message or contact the journal office at ploscompbiol@plos.org. Please include the following items when submitting your revised manuscript:

We look forward to receiving your revised manuscript.

Kind regards,

Jason Papin

Editor-in-Chief

PLOS Computational Biology

**Additional Editor Comments (if provided):**

**Journal Requirements:**

**Reviewers' comments:**

Reviewer's Responses to Questions

**Comments to the Authors:**

Reviewer #1: Xing et al. describe RETINA, a machine-learning method for segmenting ultrastructural features in electron microscopy (EM) images. Segmenting ultrastructures such as mitochondria, lysosomes, nucleoli, etc. in EM datasets is critical for their analysis since manual segmentation does not scale to large datasets. The main contribution of the authors is to implement pretraining on a related dataset into their model training to improve performance. Pretraining can leverage large amounts of unlabeled data avoiding higher costs for label generation. The authors show that pretraining improves model performance in comparison to a baseline that does not use pretraining, as well as another baseline.

The paper is written clearly, and the data is presented well. However, there are several shortcomings in the model chosen for the benchmark datasets, experiments performed and presented, the comparison models, and the metrics used.

The authors benchmark their model on several datasets that are 3d, in that they extend in all 3d dimensions. It has been long established that leveraging this 3d context improves performance, yet, the authors have chosen a model that uses 2d convolutions. This part is not entirely clear though, because the model is not fully presented in the paper; I referred to the original manuscript in which the model architecture of Transunet was presented (Chen et al., 2021). The baseline the authors compare their model to appear to be 2d models as well which do not represent the state of the art. Just as one point of reference, the CREMI leaderboard (https://cremi.org/leaderboard/) lists multiple methods applied for synapse detection. CleftNet, one of the high-scoring methods, uses a 3d residual UNet (https://arxiv.org/pdf/2101.04266). This is just one example for a stronger baseline the authors could have considered.

The authors chose to use IoU for all their benchmarks. This is a limited metrics because it does not treat voxels that separate individual objects differently from those that are merely at the edge. Yet, merging two distinct objects has a much higher impact on downstream analyses. For instance, the CREMI challenge uses an object based metric (https://cremi.org/metrics/#clefts). A side effect of only using IoU is that the presented performance cannot be compared with the performances published on the cremi challenge. If a challenge like CREMI publishes metrics, they should be used.

Still, the pretraining as implemented by the authors improves performance according to one metric (when compared to its not-pretrained self). To make the impact clear however, more evaluations and metrics should have been presented. For instance, it is not clear if the performance gap outweighs the additional cost of pretraining, etc, etc. . Overall, the impact of this work on the science that is using segmentations like the once created by RETINA is not made clear enough. The introduction claims that “deep learning-based segmentation has shown promise, its accuracy to automatically segment cellular structures in EM data remains insufficient compared to expert manual results.” There are many publications to list here that show that this is not true. Many automated methods now going back almost 10 years have shown performances similar to human annotators for ultrastructural segmentation. That is not to say that there are no improvements to be had, especially when it comes to efficiency along the lines of what RETINA tries to achieve.

Reviewer #2: Xing et al. presented a hybrid approach, RETINA, for EM segmentation. By combining the widely used TransUNet backbone with pre-trained features from the CEM500K dataset, the authors suggested that the hybrid method outperformed the other approaches in terms of speed and accuracy. It is a timely work given the rapid expansion of volume EM datasets in recent years.

However, I recommend addressing the overall accuracy of the manuscript, particularly in the following areas:

1. Could you provide the iteration time for each model? This information is essential for accurately estimating the computational resources required to achieve a certain IoU.

2. Discrepancies in the numbers reported in Table 1, Table S5, and Figure 3. For example, in line 145, it is stated that “The UroCell benchmark demonstrates the most significant IoU increase, over 150%”. I was not able to derive this number from Table 1. The Guay numbers in Table 1 do not match those in Table S5 and Figure 3. The UroCell Random Init. value of 0.203 in Table 1 appears to be incorrect. Iteration numbers of Gray, Perez, and Kasthuri++ in Figure 3 do not match with those in Table 1. The spelling of “Kathuri” in Figure 3 should be "Kasthuri".

3. The names of benchmarks in Table 1 should be better defined to ensure consistency with the main text. For example, “CEM500K moco” in Table 1 should be explicitly linked to “UNet-ResNet50 model pre-trained with MoCoV2” in line 126 and the “MoCoV2 pre-trained model” in line 158. “Random Init.” should be clearly defined as the “randomly initialized UNet-ResNet50 model” in line 131. The “non-pre-trained UNet-ResNet50 model” in line 172 is unclear and requires clarification.

4. It would be better to have the order of Benchmarks be consistent between Figure 3 and Figure S5.

**Have the authors made all data and (if applicable) computational code underlying the findings in their manuscript fully available?**

Reviewer #1: Yes

Reviewer #2: Yes

PLOS authors have the option to publish the peer review history of their article (what does this mean?). If published, this will include your full peer review and any attached files.

Reviewer #1: No

Reviewer #2: No

**Figure resubmission:**
---

## [Decision Letter · Decision Letter 1]

6 May 2025

Dear Xing,

We are pleased to inform you that your manuscript 'RETINA: Reconstruction-based Pre-Trained Enhanced TransUNet for Electron Microscopy Segmentation on the CEM500K Dataset' has been provisionally accepted for publication in PLOS Computational Biology.

Best regards,

Jason Papin

Editor-in-Chief

PLOS Computational Biology

Reviewer's Responses to Questions

**Comments to the Authors:**

Reviewer #1: The authors sufficiently addressed my points through adding 3d approaches to the evaluation, adding metrics, and clarifying the manuscript in several ways. I appreciate the extensive responses and the added work that improved the manuscript.

While I understand that the use of CEM500K is the reason for using a 2d approach in the first place, there are plenty of 3d datasets available to pretrain 3d models. Since the demonstrated performance of the RETINA exceeds that of the (reasonably) selected 3d approaches, the main point of the manuscript holds and this is left to future work.

Reviewer #2: Thank you for addressing my questions. The revised version looks good.

**Have the authors made all data and (if applicable) computational code underlying the findings in their manuscript fully available?**

Reviewer #1: Yes

Reviewer #2: Yes

PLOS authors have the option to publish the peer review history of their article (what does this mean?). If published, this will include your full peer review and any attached files.

Reviewer #1: No

Reviewer #2: No

---

## [Editor Report · Acceptance letter]

PCOMPBIOL-D-24-01938R1

RETINA: reconstruction-based pre-trained enhanced TransUNet for electron microscopy segmentation on the CEM500K dataset

Dear Dr Xing,

I am pleased to inform you that your manuscript has been formally accepted for publication in PLOS Computational Biology. Your manuscript is now with our production department and you will be notified of the publication date in due course.

With kind regards,

Anita Estes
